# The Possible Impact of Zinc-Enriched Multivitamins on Treatment-Naïve Recurrent Aphthous Stomatitis Patients

**DOI:** 10.3390/jcm14010260

**Published:** 2025-01-05

**Authors:** Hye Rin Seo, Kyung Bae Chung, Do-Young Kim

**Affiliations:** Department of Dermatology and Cutaneous Biology Research Institute, Severance Hospital, Yonsei University College of Medicine, Seoul 03722, Republic of Korea; hyerinseo17@yuhs.ac (H.R.S.); chungkyungbae@yuhs.ac (K.B.C.)

**Keywords:** stomatitis, aphthous, zinc, dietary supplements

## Abstract

**Background/Objectives:** Recurrent aphthous stomatitis is a common oral mucosal disorder characterized by painful ulcerations and frequent recurrences, which can significantly impair quality of life. This study explores the efficacy of zinc-enriched multivitamin supplementation (ZnVita, containing 22.5 mg of elemental zinc) for the treatment of recurrent aphthous stomatitis in treatment-naïve patients, aiming to diminish the reliance on immunomodulatory drugs. **Methods:** Conducted as a retrospective observational study at a tertiary referral hospital from 2013 to 2023, we analyzed 201 patients who received ZnVita daily for a minimum duration of one month as their initial management. Patients who were using systemic immunomodulating agents or met the diagnostic criteria for Behçet’s disease were excluded. **Results:** Of the 201 patients, 95 presented with an oral ulcer alone and 106 exhibited additional symptoms associated with Behçet’s disease. Efficacy analysis was conducted on 155 patients due to follow-up loss or incomplete data. Among them, 58.7% (91/155) showed partial or significant responses. Patients with BD-related symptoms were significantly more prevalent among non-responders (64.1%, 41/64) compared to responders (42.9%, 39/91), with a statistically significant difference (*p* = 0.009). Treatment was well-tolerated, with mild gastrointestinal adverse events reported in only 2.5% of cases. **Conclusions:** These results suggest that zinc-enriched multivitamin supplementation offers a beneficial and safe initial treatment alternative for a considerable proportion of treatment-naïve recurrent aphthous stomatitis patients, especially those without concurrent symptoms of Behçet’s disease, showcasing its potential in reducing the future need for immunomodulatory treatments.

## 1. Introduction

Recurrent aphthous stomatitis (RAS) is a prevalent oral mucosal disorder characterized by the episodic eruption of well-demarcated, small and rounded painful ulcers [1]. As the most prevalent disease of the oral mucosa, it affects up to 25% of the general population with individuals experiencing episodes every 1–4 months [2,3,4,5]. Although its exact etiology is unclear, factors such as genetics, immunological abnormalities, stress, local trauma, and nutritional deficiencies, including vitamin B12, vitamin D, zinc, folate, and iron, have been implicated in its pathogenesis [6].

Behçet’s disease (BD) is notably linked to recurrent aphthous stomatitis (RAS), distinguishing itself as an autoimmune multisystem disorder characterized by vasculitis in small- and medium-sized vessels. The BD Research Committee of Japan’s diagnostic criteria identify BD by major symptoms such as recurrent oral aphthous ulcers, skin lesions, ocular inflammation, and genital ulcers. Minor symptoms include arthritis, intestinal ulcers, epididymitis, vascular lesions, and neurological symptoms. Patients often do not exhibit all symptoms of BD simultaneously, with many symptoms appearing in isolation [7]. This variability makes RAS particularly challenging in diagnosing BD, especially when it manifests as the earliest feature of the disease [8]. While only a few reports have explored the distinct characteristics between RAS and BD, it is crucial to monitor patients presenting with multiple major aphthae or minor symptoms of BD closely for any progression towards BD [9].

The management of RAS remains challenging, with no definitive cure available. Current treatment strategies aim to alleviate pain, reduce the size and number of ulcers, and decrease their duration and frequency. First-line therapy often involves topical formulations possessing antiseptic, analgesic, and anti-inflammatory properties, such as chlorhexidine mouthwash, benzydamine hydrochloride, and topical corticosteroids, which are applied for the duration of the lesions’ presence. A single occurrence of ulcers may last approximately 7–14 days. For severe and constantly recurring ulcerations, where topical management proves ineffective, systemic treatments such as short-term systemic glucocorticoid, pentoxifylline (400 mg three times daily), dapsone (50–150 mg/day), and colchicine (0.5–1.5 mg/day) are considered [5]. However, usage of these medications can lead to severe side effects. Commonly used medications such as dapsone and colchicine are associated with notable adverse effects, including anemia, hemolysis, jaundice, gastrointestinal complaints, neutropenia, and liver enzyme elevation [10,11].

Recent studies highlight the role of vitamin supplementation in RAS management, particularly in individuals with specific deficiencies. Deficiencies in vitamins B1, B2, and B6 have been linked to prolonged RAS episodes, with supplementation reducing disease duration [12]. High-dose vitamin B12 has also shown efficacy in RAS treatment, independent of initial vitamin B12 blood levels [13]. However, daily multivitamin supplementation has produced inconsistent results [14], potentially due to differences in formulation, dosages, and patient characteristics.

Zinc supplementation has also gathered attention due to its anti-inflammatory and wound-healing properties [15,16]. As an essential trace element, zinc is crucial for immune function, cellular repair, and inflammation modulation [17]. Although some studies associate RAS with zinc deficiency, the relationship between zinc levels and RAS is still a subject of debate [18,19]. Clinical studies on systemic zinc supplementation in RAS patients, with daily doses ranging from 12 mg to 660 mg, have shown variable but generally positive effects with minimal side effects [19]. However, due to unspecified concurrent medication use in these studies, it is challenging to clearly understand to what extent the sole use of supplementation induces clinical improvement in patients with RAS.

In this context, our study investigates the efficacy and safety of zinc-enriched multivitamin formulation in treatment-naïve RAS patients in a real-world setting. Practical information on multivitamin supplementation, commonly used for its safety and accessibility, will provide a comprehensive management strategy for RAS, especially for patients who have not undergone previous treatments, potentially sparing them from the use of immunomodulatory agents with severe adverse events.

## 2. Materials and Methods

This retrospective observational study received approval from the Institutional Review Board (IRB) at a tertiary referral hospital in Seoul, Republic of Korea (approval number: 4-2023-1685). It reviewed patients diagnosed with RAS at the dermatology outpatient clinic between March 2013 and February 2023, with clinical data collection continuing until February 2024.

In line with our institution’s standard clinical practice, individuals who had not received any systemic medication for oral ulcers in the past year and were diagnosed with RAS were routinely prescribed a zinc-enriched multivitamin supplement (ZnVita) as part of their initial management. By leveraging this predefined clinical practice pattern, we were able to utilize a cohort that exclusively experienced exposure to ZnVita, thereby facilitating an evaluation of the drug response.

The inclusion criteria encompassed individuals over 18 years diagnosed with idiopathic RAS or suspected BD (based on diagnostic criteria recommended by the BD research committee of Japan [20]). Patients were identified through the clinical data repository system, with suspected BD characterized by a combination of RAS and genital ulcers or inflammatory skin conditions, such as erythema nodosum, excluding complete or incomplete BD [20]. Accordingly, patients were excluded if they exhibited three or more BD-related symptoms, characteristic ocular involvement, or at least two minor symptoms as defined by the BD Research Committee of Japan, which would satisfy the diagnostic criteria for BD. Additionally, patients were excluded if they had been using immunomodulatory agents (e.g., colchicine, steroids, dapsone, thalidomide, or TNF-alpha inhibitors) or had concurrent systemic conditions like syphilis, inflammatory bowel disease, or autoimmune disorders. This approach ensured that the study population consisted of individuals receiving ZnVita as their sole therapeutic intervention for RAS, without the influence of other systemic treatments.

ZnVita (Zeten C^TM^, Hanmi Pharm., Seoul, Republic of Korea) contains 22.5 mg of elemental zinc along with other multivitamin components, including 750 mg of ascorbic acid, 20 mg of calcium pantothenate, 12 mg of cyanocobalamin powder 0.1%, 0.4 mg of folic acid, 100 mg of nicotinamide, 20 mg of pyridoxine hydrochloride, 15 mg of riboflavin, 15 mg of thiamine nitrate, and 60 mg of tocopherol acetate powder 50%. The regimen for ZnVita was one tablet daily.

The demographic characteristics (sex, age at disease presentation), clinical characteristics (types of oral ulcers, duration of disease, BD-related symptoms, duration of ZnVita, duration of observation), and laboratory features (blood measurements, HLA-B51 genotype status, erythrocyte sedimentation rate (ESR), C-reactive protein (CRP)) of the patients were obtained from their medical records. The “duration of disease” was defined as the time elapsed from the onset of recurrent aphthous stomatitis (RAS) symptoms to the initiation of ZnVita treatment. The “duration of observation” refers to the period from the start of zinc supplementation to either loss of follow-up or the end of the study. The “duration of ZnVita” denoted the period during which the patients were taking ZnVita treatment.

Patients were assessed for therapeutic efficacy based on at least two follow-up visits after initiating ZnVita treatment, with a minimum adherence of 1 month. The outcomes were stratified into three tiers based on patient-reported outcomes, which encompassed assessments of oral ulcer activity, pain status, and functional states. The functional state was evaluated according to how oral ulcers affected daily activities, including experiencing an unpleasant taste and difficulties with speaking, eating, chewing, and swallowing. Patient-reported outcomes were assessed using a pre-defined, structured follow-up form, which categorized treatment responses into three groups: significant improvement, partial improvement, or no improvement. A ‘significant response’ was assigned to patients who demonstrated improvements across all the aforementioned categories after ZnVita treatment and did not require any additional immunomodulatory agents during the observation period. Patients exhibiting improvements in one or more, but not all, specified categories were classified as having a ‘partial response’. Those who did not show any improvement in these measures were categorized as having ‘no response’. The ‘responder’ group included patients categorized as either significant or partial responders, whereas the ‘non-responder’ group consisted of patients who showed no improvement.

Continuous data were presented as mean value ± standard deviation (SD) or median with interquartile range. Categorical data were presented as a frequency of certain categories. A chi-square test was used to assess the significance of differences in frequency distributions. The differences between the two groups were assessed using Student’s t-test for variables with a normal distribution. A *p*-value of less than 0.05 was considered statistically significant. The association between responder status and each variable was evaluated using binomial logistic regression analyses. Multicollinearity was checked for the data (variance inflation factor <10). The model fitness was checked using Hosmer and Leme to show the goodness of fit (*p* > 0.05). Data analysis was performed using SPSS Statistics 21 (Chicago, IL, USA).

## 3. Results

A total of 201 patients with idiopathic RAS treated with ZnVita supplementation were included in the study. A summary of patient demographics and characteristics is presented in Table 1. The mean age of the participants was 41.3 ± 14.4 years. Consistent with epidemiological trends, a majority of the patients were female, accounting for 65.7% (132/201) of the study population [8]. The duration of disease prior to the study varied among patients, with a mean of 9.5 years. Minor aphthous ulcers were the most common presentation, found in 74.3% (110/148) of the patients. Among the patients, 106 patients (52.7%) exhibited additional symptoms associated with BD, such as episodes of genital ulcers or inflammatory skin conditions like erythema nodosum, along with RAS. However, none of these patients met the diagnostic criteria for BD during the observation period. The mean duration of observation was 562 days, while the mean duration of ZnVita was 307.5 days. The mean duration of observation was longer than the mean duration of ZnVita because some patients, particularly those with lesser responses to ZnVita, changed to other immunomodulatory drugs during the follow-up period. This led to shorter zinc supplementation periods compared to the overall observation days.

Of the 201 patients, 155 were available for efficacy analysis, after excluding 31 due to loss to follow-up and 15 for incomplete documentation, as shown in Figure 1. Among the evaluated patients, 91 patients (58.7%) reported improvements in patient-reported outcomes related to oral ulcer activity, pain status, and functional states. Specifically, 25 patients (16.1%) showed improvement across all categories, without the need for additional immunomodulatory agents. A subgroup analysis of the 75 patients presenting solely with RAS symptoms showed that 52 patients (69.3%) reported improvements in patient-reported outcomes. In contrast, among 80 patients with additional BD-related symptoms, only 39 patients (48.8%) reported similar levels of improvement, underscoring the potential impact of overlapping BD symptoms on treatment efficacy with ZnVita (Table 2 and Figure 1).

The comparative analysis between non-responders (n = 64) and responders (n = 91) revealed no significant differences in age at presentation, sex, or duration of disease. However, BD-related symptoms were significantly more prevalent among non-responders, with 64.1% (41/64) of non-responders presenting such symptoms compared to 42.9% (39/91) of responders (*p*-value = 0.009), as detailed in Table 3. Further analysis through logistic regression, which included only patients with complete variable documentation (n = 134), indicated that the presence of other BD-related symptoms was associated with a lower likelihood of responding to ZnVita (odds ratio, 0.435; 95% CI, 0.195–0.992, *p* = 0.042) (Appendix A).

Laboratory parameters, including blood cell count ratios (neutrophil-to-lymphocyte ratio (NLR), monocyte-to-lymphocyte ratio (MLR), platelet-to-lymphocyte ratio (PLR) and mean platelet volume-to-platelet count (MPV/PC) ratio), ESR, CRP, and HLA-B51 genotype, did not significantly differ between groups, though non-responders showed marginally higher ESR and CRP levels (Table 4). The overall safety profile for ZnVita supplementation was favorable. Only five patients (2.5%) experienced mild gastrointestinal adverse events, leading to discontinuation in four cases.

## 4. Discussion

Our study provides comprehensive data on the demographics and laboratory features of treatment-naïve patients diagnosed with idiopathic RAS and treated with ZnVita supplementation within a substantial real-world cohort. Notably, approximately half of the patients demonstrated a partial or significant response to ZnVita treatment, indicating that zinc-enriched multivitamin supplementation could serve as a viable therapeutic option for treatment-naïve RAS patients. A total of 20% of patients presenting with RAS symptoms alone showed significant improvement, potentially reducing the need for immunomodulatory agents.

Despite the lack of significant demographic differences between responders and non-responders, our analysis revealed a higher prevalence of BD-related symptoms among non-responders. Furthermore, logistic regression analysis showed that the presence of other BD-related symptoms was linked to a decreased likelihood of a positive response to ZnVita treatment. These results suggest that the additional inflammatory burden in patients with additional BD-related symptoms may necessitate further immune modulation for clinical improvement.

Zinc supplementation plays a critical role in modulating immune responses and promoting tissue repair, which could explain its beneficial effects in managing RAS and BD [17,21]. Zinc is essential for regulating pro-inflammatory cytokines, such as TNF-α and IL-1β, which are implicated in the pathogenesis of RAS [22]. It also stabilizes cell membranes and enhances wound healing through its involvement in epithelial repair and angiogenesis [15]. These mechanisms likely contribute to the observed improvements in oral ulcer activity and pain reduction in patients treated with zinc-enriched multivitamins. Furthermore, the therapeutic potential of zinc supplementation in reducing the inflammatory burden and enhancing mucosal healing underscores its suitability as an initial treatment for RAS, especially for patients without BD-related symptoms.

Both ESR and CRP are well-known laboratory markers reflecting systemic inflammation. In addition, the neutrophil-to-lymphocyte ratio (NLR), monocyte-to-lymphocyte ratio (MLR), platelet-to-lymphocyte ratio (PLR), and mean platelet volume-to-platelet count (MPV/PC) ratio have been investigated as cost-effective circulating clinical markers of chronic, low-grade inflammation in many diseases [23,24]. While we observed no significant differences in inflammatory markers between responders and non-responders, the absence of baseline zinc status assessments limits a definitive conclusion about the impact of zinc supplementation based on these markers. The absence of these data notwithstanding, zinc’s established anti-inflammatory and wound-healing properties [17] suggest its potential role in the noted clinical improvement, despite not being reflected in laboratory markers. Future research, including initial evaluations of zinc levels, will be vital to further explore this relationship and provide a more comprehensive understanding of zinc’s role in modulating inflammation.

Zinc supplementation is associated with a low risk profile, and toxicity typically occurs at substantially higher intake levels (225–450 mg/day) compared to the commonly consumed amounts ranging from 15 to 100 mg Zn per day [25]. The dose used in our study (22.5 mg of elemental zinc per day) was in the range of the standard daily recommendation (8–35 mg per day), and adverse events in our cohort were infrequent and mild, primarily consisting of gastrointestinal complaints such as nausea and abdominal discomfort, which led to treatment discontinuation in four cases. Despite this favorable safety profile, the relatively high overall discontinuation rate observed may reflect the challenges of adherence to supplementation regimens, compounded by the availability of alternative treatments for RAS.

The long-term use of zinc-enriched multivitamins as a first-line treatment for RAS may offer potential benefits, including sustained anti-inflammatory effects and enhanced mucosal healing. However, prolonged supplementation could carry risks such as copper deficiency or gastrointestinal issues, particularly if not properly monitored [26]. To validate these findings, future research should include randomized controlled trials or prospective studies that assess the long-term safety and efficacy of zinc supplementation in RAS, with a focus on identifying optimal dosage, treatment duration, and potential adverse effects.

This study has certain limitations. The exclusion of a considerable number of patients due to loss of follow-up or inadequate documentation may introduce selection bias. Additionally, the retrospective nature of this study, coupled with our institution’s standard clinical practice of prescribing ZnVita to all treatment-naïve patients presenting with RAS, inherently limited our ability to include a comparator group. Furthermore, the absence of initial zinc blood level assessments precludes the evaluation of its influence on treatment response. Moreover, the challenges in quantifying nutritional status with objective and reliable markers necessitated the exclusion of this variable from our study, acknowledging its potential impact on the intervention’s effectiveness.

## 5. Conclusions

This study demonstrated that zinc-enriched multivitamin supplementation is generally well tolerated and beneficial for approximately half of the treatment-naïve RAS patients, with 16.1% of them avoiding further immunomodulatory treatment. Patients with RAS alone responded more favorably (69.3%) compared to those with Behçet’s disease-related symptoms (48.8%). The treatment showed minimal adverse events, with only 2.5% of patients experiencing mild gastrointestinal side effects. We advocate for the consideration of zinc-enriched multivitamin as an initial therapy for treatment-naïve RAS patients, particularly those without concomitant BD symptoms. While our study did not identify predictive laboratory markers, further research is warranted to identify the potential confounders and predictors for treatment response in idiopathic RAS.

## Figures and Tables

**Figure 1 jcm-14-00260-f001:**
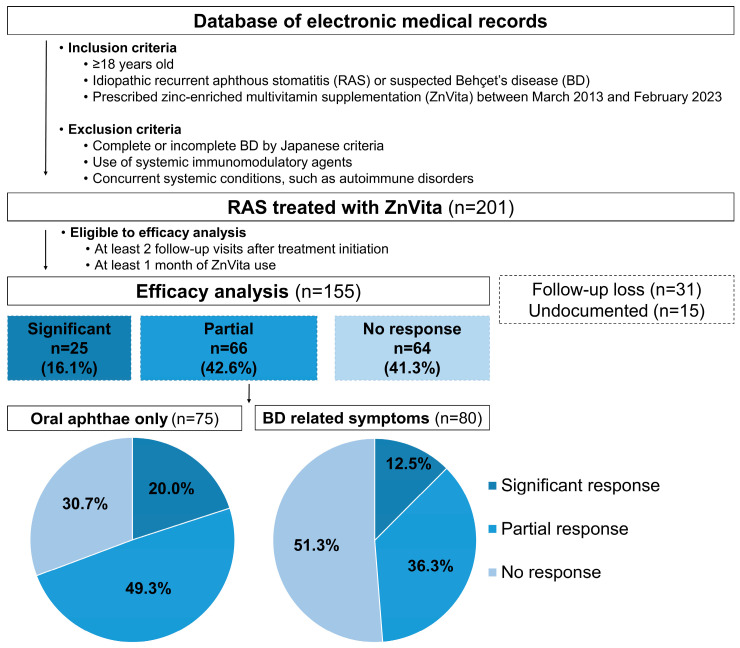
Flowchart of patient enrolment and efficacy analysis in the ZnVita study.

**Table 1 jcm-14-00260-t001:** Baseline characteristics of patients given ZnVita for recurrent aphthous stomatitis.

Characteristic	Patients
	(N = 201)
Age at presentation, year	
Mean (range)	41.3 (30.2–52.5)
Female sex	132 (65.7%)
Duration of disease, year (n = 192) *	
Mean (range)	9.5 (3.0–15.0)
Oral ulcer type (n = 148) *	
Minor aphthae only	110 (74.3%)
Major aphthae only	9 (6.1%)
Herpetiform only	3 (2.0%)
Mixed type	26 (17.6%)
Duration of observation, days	
Mean (range)	562.0 (51.0–665.0)
Duration of ZnVita, days	
Mean (range)	307.5 (41.5–299.0)
Associated symptom	
Oral aphthae only ^†^	95 (47.3%)
Behçet’s disease-related symptoms ^‡^	106 (52.7%)

* Exclusion of undocumented data. ^†^ ‘Oral aphthae only’ indicated patients with recurrent aphthous stomatitis without any other symptoms that meet the major criteria for Behçet’s disease. ^‡^ ‘Behçet’s disease-related symptoms’ refer to patients with recurrent aphthous stomatitis and one or more additional symptoms associated with BD, without meeting diagnostic criteria of complete or incomplete BD during the observation period.

**Table 2 jcm-14-00260-t002:** Therapeutic response to ZnVita supplementation in idiopathic RAS patients.

Therapeutic Response	Total Patients * (N = 155)	Associated Symptoms	
		Oral Aphthae Only (N = 75)	Behçet’s Disease-Related Symptoms (N = 80)
No response	64 (41.3)	23 (30.7)	41 (51.3)
Any improvement	91 (58.7)	52 (69.3)	39 (48.8)
Partial response	66 (42.6)	37 (49.3)	29 (36.3)
Significant response	25 (16.1)	15 (20.0)	10 (12.5)

Data are shown as number (%). * Of the 201 patients treated with ZnVita, 155 were assessed for therapeutic efficacy. Data on follow-up were not available for 47 patients due to loss to follow-up (n = 31) or undocumented cases (n = 15).

**Table 3 jcm-14-00260-t003:** Comparison of demographic and clinical characteristics among ZnVita responders and non-responders in RAS patients.

		Non-Responder (N = 64)	Responder (N = 91)	Total (N = 155)	*p*-Value
Age at presentation, year		40.1 ± 13.5	42.0 ± 15.1	41.2 ± 14.5	0.08
Duration of disease, year, (n = 148) *		9.0 ± 9.1	9.9 ± 8.5	9.6 ± 8.7	0.709
Sex					0.524
	Male	25/64 (39.1)	31/91 (34.1)	56/155 (36.1)	
	Female	39/64 (60.9)	60/91 (65.9)	99/155 (63.9)	
Associated symptom					0.009 ^†^
	Oral aphthae only	23/64 (35.9)	52/91 (57.1)	75/155 (48.4)	
	Behçet’s disease-related symptoms	41/64 (64.1)	39/91 (42.9)	80/155 (51.6)	

Data are shown as number (%), mean ± SD (standard deviation). * Exclusion of undocumented data. Significant differences are marked by ^†^ (*p* < 0 01).

**Table 4 jcm-14-00260-t004:** Laboratory parameter comparison by ZnVita response in RAS patients.

	Non-Responder (N = 64)	Responder (N = 91)	Total (N = 155)	*p*-Value
NLR * (n = 148)	2.0 ± 0.8	2.0 ± 0.8	2.0 ± 0.8	0.436
MLR * (n = 148)	0.2 ± 0.1	0.2 ± 0.1	0.2 ± 0.1	0.886
PLR * (n = 148)	133.6 ± 48.8	143.7 ± 64.1	139.6 ± 58.5	0.514
MPV/PC * (n = 148) (fL 10^−5^ μL^−1^)	0.04 ± 0.01	0.03 ± 0.01	0.03 ± 0.01	0.807
ESR * (n = 145)	20.6 ± 16.4	19.5 ± 14.4	19.9 ± 15.2	0.455
CRP * (n = 145)	4.5 ± 19.4	2.3 ± 8.7	3.2 ± 13.9	0.142
HLA-B51 genotype * (n = 150)				0.431
Positive	16/60 (26.7)	19/90 (21.1)	35/150 (23.3)	
Negative	44/60 (73.3)	71/90 (78.9)	115/150 (76.7)	

Data are shown as number (%), mean ± SD (standard deviation). Abbreviations; NLR, neutrophil-to-lymphocyte ratio; MLR, monocyte-to-lymphocyte ratio; PLR, platelet-to-lymphocyte ratio; MPV, mean platelet volume; PC, platelet count; ESR, Erythrocyte sedimentation rate; CRP, C-related protein. * Exclusion of undocumented data.

## Data Availability

The data supporting the findings of this study are available from the corresponding author upon reasonable request.

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
