# Peer review of "The Possible Impact of Zinc-Enriched Multivitamins on Treatment-Naïve Recurrent Aphthous Stomatitis Patients"

_jcm, 2025, doi:10.3390/jcm14010260_

Round 1

Reviewer 1 Report

Comments and Suggestions for Authors

Dear authors,

thank you for the opportunity to revise this manuscript.

I recommend the following revisions.

Abstract

Please reduce ‘Results’  

Introduction

-       “However, usage of … different patients characteristics” Please summarize the results of these studies, removing the authors' names and creating a unified statement 

-       “Halboub et al. … with minimal side effects” Please summarize the results of these studies, removing the authors' names and creating a unified statement 

M&M

Well done 

Results  

-       Table 1: please reduce the text in favor of Tables to make the manuscript more readable and make the table more readable by better separating the different characteristics listed 

-       Figure 1: please add the reference for the flowchart provided

Discussion

Please expand the discussion, as it is too short

Conclusions

Please add a few lines from the missing 'Conclusions' paragraph

Author Response

Comment 1: Abstract: Please reduce ‘Results’  

Response 1: We have streamlined the 'Results' section in the Abstract by condensing the details while retaining the essential findings. The revised Abstract now provides a concise summary of the key outcomes.

Of the 201 patients, 95 presented oral ulcer alone and 106 exhibited additional symptoms associ-ated with Behçet’s disease. Efficacy analysis was conducted on 155 patients due to follow-up loss or incomplete data. Among them, 58.7% (91/155) showed partial or significant responses. Pa-tients with BD-related symptoms were significantly more prevalent among non-responders (64.1%, 41/64) compared to responders (42.9%, 39/91), with a statistically significant difference (p = 0.009). Treatment was well-tolerated, with mild gastrointestinal adverse events reported in only 2.5% of cases.

Comment 2: in Introduction

-      “However, usage of … different patients characteristics” Please summarize the results of these studies, removing the authors' names and creating a unified statement 

-       “Halboub et al. … with minimal side effects” Please summarize the results of these studies, removing the authors' names and creating a unified statement 

Response 2: We have revised this section by summarizing the results of the referenced studies into a unified statement, removing the individual authors' names.

Deficiencies in vitamins B1, B2, and B6 have been linked to prolonged RAS episodes, with supplementation reducing disease duration [12]. High-dose vitamin B12 has also shown efficacy in RAS treatment, independent of initial vitamin B12 blood levels [13]. However, daily multivitamin supplementation has produced inconsistent results [14], potentially due to differences in formulation, dosages, and patient characteristics. ….. Clinical studies on systemic zinc supplementation in RAS patients, with daily doses ranging from 12 mg to 660 mg, have shown variable but generally positive effects with minimal side effects [17].

Comment 3. In Results  

-       Table 1: please reduce the text in favor of Tables to make the manuscript more readable and make the table more readable by better separating the different characteristics listed 

-       Figure 1: please add the reference for the flowchart provided.

Response 3: We appreciate the reviewer’s insightful suggestion regarding Table 1. In response, we have restructured the table to improve readability and conciseness. Specifically, we have streamlined the presentation of data by combining the mean (SD) and median (range) into a single format as mean (range), which we believe enhances clarity while retaining the essential details.  

Figure 1. Patient enrollment and efficacy analysis flowchart. This flowchart was developed based on the study methodology described in the Materials and Methods section. While the flowchart may appear conceptually similar to plots from prior studies in the same field, it reflects the unique design and results of our current research. Therefore, we believe that direct citation of prior works is not warranted in this context.

Comment 4: Discussion: Please expand the discussion, as it is too short

Response 4: We have expanded the Discussion section to reinforce the mechanistic background of zinc supplementation in RAS. The added content elaborates on zinc’s role in modulating pro-inflammatory cytokines, stabilizing cell membranes, and promoting epithelial repair, providing a stronger theoretical foundation for its therapeutic effects. To substantiate these discussions, we have included three additional references that highlight zinc's established roles in inflammation modulation and tissue repair.

Comment 5: Conclusions: Please add a few lines from the missing 'Conclusions' paragraph.

Response 5: The revised Conclusions section reorganizes and condenses key points from the Discussion to provide a concise summary of our findings. This adjustment ensures that the practical implications and clinical relevance of zinc-enriched multivitamin supplementation for treatment-naïve RAS patients are clearly articulated.

The study demonstrated that zinc-enriched multivitamin supplementation is gener-ally well tolerated and beneficial for approximately half of the treatment-naïve RAS pa-tients, with 16.1% of them avoiding further immunomodulatory treatment. Patients with RAS alone responded more favorably (69.3%) compared to those with Behçet’s dis-ease-related symptoms (48.8%). The treatment showed minimal adverse events, with only 2.5% of patients experiencing mild gastrointestinal side effects. We advocate for the con-sideration of zinc-enriched multivitamin as an initial therapy for treatment-naïve RAS pa-tients, particularly those without concomitant BD symptoms. While our study did not identify predictive laboratory markers, further research is warranted to identify the poten-tial confounders and predictors for treatment response in idiopathic RAS.

Reviewer 2 Report

Comments and Suggestions for Authors

Dear Authors,

The article  entitled  Efficacy of Zinc-Enriched Multivitamin in Treatment-Naïve Recurrent Aphthous Stomatitis Patients” presents a topic of great importance with a profound impact on the specialized medical practice.

I have some remarks for the authors:

·         In the Abstract a short introduction is missing.

·         In the Abstract, in Results will not be used abbreviations as:RAS”, “BD”.

·         In Introduction, the authors mentioned: “First-line therapy often involves topical formulations possessing antiseptic, analgesic, and anti-inflammatory properties, which are applied for the duration of the lesions’ presence.”………… Here it is necessary to give some examples.

·         In Introduction, the authors mentioned: “For severe and constantly recurring ulcerations, where topical management proves ineffective, systemic treatments such as short-term systemic glucocorticoid, pentoxifylline, dapsone and colchicine are considered [5].”………… Here it is necessary to mention the therapeutic schemes.

·         In Introduction, the authors mentioned: “However, usage of these medications can lead to severe side effects that limit their utility.”………Here is to mention only “However, usage of these medications can lead to severe side effects”…………….do not use “that limit their utility”.

·         In 2. Materials and Methods, the authors mentioned: “In line with our institution's standard clinical practice, individuals who had not received any systemic medication for oral ulcers in the past year and were diagnosed with RAS were routinely prescribed a zinc-enriched multivitamin supplement (ZnVita) as part of their initial management.” …………………….It is necessary to mention the producer for the zinc-enriched multivitamin supplement (ZnVita).

·         In Discussion, the authors mentioned: “Dose used in our study (22.5 mg of elemental zinc per day) was in the range of standard daily recommendation (8 – 35 mg per day), and adverse events in our cohort were infrequent and mild.”  It is necessary to specify the adverse events encountered in the present study.

·         The title it is necessary to be modified !  As the authors mentioned in Discussion: “….the absence of  initial zinc blood level assessments precludes the evaluation of its influence on treatment response.”   ……..AS A SUGGESTED TITLE :“The Possible Impact of Zinc-Enriched Multivitamin in Treatment-Naïve Recurrent Aphthous Stomatitis Patients”

Author Response

Comment 1:    In the Abstract a short introduction is missing.

Response 1: We appreciate the reviewer’s suggestion. A brief introductory sentence has been added to the Abstract to provide context for the study. The revised Abstract begins with the sentence:

"Recurrent aphthous stomatitis is a common oral mucosal disorder characterized by painful ulcerations and frequent recurrences, which can significantly impair quality of life."

Comment 2:    In the Abstract, in Results will not be used abbreviations as: “RAS”, “BD”.

Response 2: Thank you for pointing this out. We have revised the Abstract to replace abbreviations with full terms.

Comment 3:   In Introduction, the authors mentioned: “First-line therapy often involves topical formulations possessing antiseptic, analgesic, and anti-inflammatory properties, which are applied for the duration of the lesions’ presence.”………… Here it is necessary to give some examples.

Response 3: We have added examples of topical formulations to this sentence in the Introduction. The revised sentence now reads: “First-line therapy often involves topical formulations possessing antiseptic, analgesic, and anti-inflammatory properties, such as chlorhexidine mouthwash, benzydamine hydrochloride, and topical corticosteroids, which are applied for the duration of the lesions’ presence.”

Comment 4:   In Introduction, the authors mentioned: “For severe and constantly recurring ulcerations, where topical management proves ineffective, systemic treatments such as short-term systemic glucocorticoid, pentoxifylline, dapsone and colchicine are considered [5].”………… Here it is necessary to mention the therapeutic schemes.

Response 4: We have included therapeutic schemes for the systemic treatments mentioned. The revised text now reads: “…. , systemic treatments such as short-term systemic glucocorticoid, pentoxifylline (400 mg three times daily), dapsone (50–150 mg/day), and colchicine (0.5–1.5 mg/day) are considered [5].

Comment 5:   In Introduction, the authors mentioned: “However, usage of these medications can lead to severe side effects that limit their utility.”………Here is to mention only “However, usage of these medications can lead to severe side effects”…………….do not use “that limit their utility”.

Response 5: We have revised this sentence as suggested. The text now reads: “However, usage of these medications can lead to severe side effects.”

Comment 6: In 2. Materials and Methods, the authors mentioned: “In line with our institution's standard clinical practice, individuals who had not received any systemic medication for oral ulcers in the past year and were diagnosed with RAS were routinely prescribed a zinc-enriched multivitamin supplement (ZnVita) as part of their initial management.” …………………….It is necessary to mention the producer for the zinc-enriched multivitamin supplement (ZnVita).

Response 6: We have added the producer information for ZnVita as “(Zeten CTM, Hanmi Pharm., Seoul, Korea)”. If including the specific product name (Zeten C) is against the journal's policy, we are open to removing it.

Comment 7:   In Discussion, the authors mentioned: “Dose used in our study (22.5 mg of elemental zinc per day) was in the range of standard daily recommendation (8 – 35 mg per day), and adverse events in our cohort were infrequent and mild.”  It is necessary to specify the adverse events encountered in the present study.

Response 7: We have specified the adverse events encountered in the present study in the Discussion section. The revised text now reads: “… mild, primarily consisting of gastrointestinal complaints such as nausea and abdominal discomfort, which led to treatment discontinuation in four cases.”

Comment 8:   The title it is necessary to be modified !  As the authors mentioned in Discussion: “….the absence of  initial zinc blood level assessments precludes the evaluation of its influence on treatment response.”   ……..AS A SUGGESTED TITLE :“The Possible Impact of Zinc-Enriched Multivitamin in Treatment-Naïve Recurrent Aphthous Stomatitis Patients”

Response 8: We agree with the reviewer’s suggestion to make the title more reflective of the study's scope and limitations. The title has been revised to: "The Possible Impact of Zinc-Enriched Multivitamin in Treatment-Naïve Recurrent Aphthous Stomatitis Patients."

Reviewer 3 Report

Comments and Suggestions for Authors

The paper is a retrospective observational study investigating the efficacy of zinc-enriched multivitamin supplementation (ZnVita) in treating recurrent aphthous stomatitis (RAS) in treatment-naïve patients. The study analyzed 201 patients who received ZnVita daily for at least one month, focusing on patient-reported outcomes including oral ulcer activity, pain status, and functional states.

My comments:

Abstract:

Include specific p-values for the main findings.

Introduction:

Provide more recent references to support the rationale for using zinc-enriched multivitamins.

Methods:

The authors should provide more details on the exclusion criteria for Behçet's disease and explain how patient-reported outcomes were measured and quantified.

Discussion:

Please, discuss the long-term implications of using zinc-enriched multivitamins as a first-line treatment for RAS, including any potential risks or benefits of prolonged use.

An interesting added point is to propose specific next steps for research, including potential randomized controlled trials or prospective studies to further validate these findings.

Author Response

Comment 1: Abstract: Include specific p-values for the main findings.

Response 1: We appreciate the reviewer’s suggestion. Specific p-values have been added to the Abstract to enhance the clarity and statistical rigor of the presented findings. 

Patients with BD-related symptoms were significantly more prevalent among non-responders (64.1%, 41/64) compared to responders (42.9%, 39/91), with a statistically significant difference (p = 0.009).

Comment 2: Introduction: Provide more recent references to support the rationale for using zinc-enriched multivitamins.

Response 2: Thank you for this suggestion. We have updated the Introduction to include more recent references supporting the rationale for using zinc-enriched multivitamins in the management of RAS. While we did not find recent studies directly related to mucosal healing, we added relevant research on wound healing (Lin et al., 2017) and T-cell response (Nikoonezhad, 2024), which we believe are appropriate and pertinent to the context.

Comment 3: Methods:The authors should provide more details on the exclusion criteria for Behçet's disease and explain how patient-reported outcomes were measured and quantified.

Response 3: We have revised the Methods section to provide additional details on the exclusion criteria for Behçet's disease. Specifically: “Accordingly, patients were excluded if they exhibited three or more BD-related symptoms, characteristic ocular involvement, or at least two minor symptoms as defined by the BD Research Committee of Japan, which would satisfy the diagnostic criteria for BD.”

For outcome measurement, in our institution, patient follow-up forms are pre-defined and structured to categorize treatment responses into three categories: significant improvement, partial improvement, or no improvement. These criteria, as described in the Methods section, were used consistently during structured interviews to assess outcomes. Added sentence for methods section: “Patient-reported outcomes were assessed using a pre-defined, structured follow-up form, which categorized treatment responses into three groups: significant improvement, partial improvement, or no improvement.” However, the three predefined categories used for assessment were not based on precise quantitative criteria but rather represent a lumped classification, reflecting the inherent limitations of this retrospective study design.

Comment 4: Discussion:

Please, discuss the long-term implications of using zinc-enriched multivitamins as a first-line treatment for RAS, including any potential risks or benefits of prolonged use.

An interesting added point is to propose specific next steps for research, including potential randomized controlled trials or prospective studies to further validate these findings.

Response 4: We have expanded the Discussion to address the long-term implications of using zinc-enriched multivitamins. The revised section discusses potential benefits, such as sustained anti-inflammatory effects and improved mucosal healing, alongside risks like copper deficiency and gastrointestinal issues associated with prolonged zinc use. Furthermore, we propose specific next steps for research: “The long-term use of zinc-enriched multivitamins as a first-line treatment for RAS may offer potential benefits, including sustained anti-inflammatory effects and enhanced mucosal healing. However, prolonged supplementation could carry risks such as copper deficiency or gastrointestinal issues, particularly if not properly monitored [26]. To validate these findings, future research should include randomized controlled trials or pro-spective studies that assess the long-term safety and efficacy of zinc supplementation in RAS, with a focus on identifying optimal dosage, treatment duration, and potential adverse effects.

Round 2

Reviewer 2 Report

Comments and Suggestions for Authors

The article can be published in this latest, revised, version.

Reviewer 3 Report

Comments and Suggestions for Authors

No more comments.